# Risk Prediction for Winter Road Accidents on Expressways

**Daeseong Kim [1], Sangyun Jung [2] and Sanghoo Yoon [2,*]**

1 Department of Statistics, Daegu University, Gyeongsan 38453, Korea; dlwnsh455@daegu.ac.kr
2 Division of Mathematics and Big Data Science, Daegu University, Gyeongsan 38453, Korea; dfsang2532@daegu.ac.kr
* Correspondence: statstar@daegu.ac.kr

**Abstract:** Road accidents caused by weather conditions in winter lead to higher mortality rates than in other seasons. The main causes of road accidents include human carelessness, vehicle defects, road conditions, and weather factors. If the risk of road accidents with changes in road weather conditions can be quantitatively evaluated, it will contribute to reducing the road accident fatalities. The road accident data used in this study were obtained for the period 2017 to 2019. Spatial interpolation estimated the weather information; geographic information system (GIS) and Shuttle Radar Topography Mission (SRTM) data identified road geometry and accident area altitude; synthetic minority oversampling technique (SMOTE) addressed the data imbalance problem between road accidents due to weather conditions and from other causes, and finally, machine learning was performed on the data using various models such as random forest, XGBoost, neural network, and logistic regression. The training- to test data ratio was 7:3. Random forest model exhibited the best classification performance for road accident status according to weather risks. Thus, by applying weather data and road geometry to machine learning models, the risk of road accidents due to weather conditions in the winter season can be predicted and provided as a service.

**Keywords:** machine learning; random forest; spatial interpolation; SMOTE; traffic safety service

## 1. Introduction

The ever-increasing vehicular traffic has resulted in corresponding increase in fatalities due to traffic accidents [1]. In particular, road accidents on expressways are reported to have higher mortality rates than those on other types of roads. Specifically, frozen or snow-covered roads are primarily responsible for fatal and large-scale accidents [2]. During the winter season in the Korean peninsula (with four distinct seasons), the roads are covered with a large amount of ice. In this regard, a service that presents prediction and guidance on the risks of traffic accidents due to weather conditions in winter may reduce the road accidents fatalities.

The identification of causes of road accidents is one of the main goals of road accident analysis [3]. The major causes of road accidents include human carelessness, vehicle defects, road conditions, and weather variables. In particular, changes in snowfall, rainfall, and weather have a significant impact on road safety, by reducing the driver's visibility, and the friction between the vehicle and the road [4]. Among these causes, factors such as human carelessness and vehicle defects occur by chance, making it difficult to predict and provide guidance in advance. However, the quantitative prediction of the risk of road accidents due to weather conditions is possible using weather information and road geometry. In this study, we aim to investigate the causes of past traffic accidents on highways in winter and examine the relationship between factors affecting accidents, such as road weather conditions and road geometry using various machine learning models.

There has been an active research on traffic accidents and prediction models using a wide range of approaches. Venkat predicted the percentage of road accidents and determining factors of accidents using the random forest model, logistic regression model, decision

tree model and k-nearest neighbor algorithms [5]. Paul et al. performed simulations on the prediction of road accidents and their severity using the decision tree, random forest, multilayer perceptron, and naive Bayes models [6]. Schlogl examined the causes for road accidents using the random forest model and the XGBoost model. The factors considered in the study include traffic volume, road geometry, road surface conditions, and weather information [7]. Eboli et al. examined factors influencing accident severity and occurrence using a logistic regression model [8]. Studies on the factors that cause road accidents include, the investigation of the effects of road geometry and weather conditions [9], traffic volume and road friction force in winter [10], and road accident risk owing to weather effects [11]. In this study, factors such as road weather, surrounding altitude, presence of bridges and tunnels, turning radius, and angle of rotation were applied to various machine learning models to predict general road accidents and risks due to weather, and key factors were determined.

The Korean Ministry of Land, Infrastructure and Transport (MOLIT) discloses information on road accident occurrence locations with the classification of construction/accident/ weather through MOLIT OpenAPI Service [12]. "Construction" provides information on areas where traffic control has been applied due to construction work, "Accident" provides information on road accident points, and "Weather" provides information on the location of road accidents due to weather. However, this service does not provide information other than the location of the road accident, the type of accident, and the time of the accident.

The Korea Meteorological Administration (KMA) discloses weather information observed through an automatic weather station (AWS) and an automated synoptic observing system (ASOS) through the KMA website [13]. However, these weather stations are designed to monitor weather changes in the Korean Peninsula and do not directly collect road-level weather information.

In this study, data from the winter season (December, January, and February) for the period from 2017 to 2020 were used for the AWS, and ASOS data provided by the KMA were used for the interpolation of weather information at road accident points. According to Kim et al., the random forest model is the optimal model for temperature and precipitation, and the generalized additive model is optimal for humidity and air pressure for interpolating road-level weather information [14]. The weather information was interpolated using optimal spatial interpolation based on the time of the accident and the location information for use in this study. For road geometry, information on the angle of rotation and turning radius was generated using GIS information of standard node links developed by MOLIT and the Douglas-Peucker algorithm.

In this study, a combined database was constructed using road accident data and standard node links of MOLIT, weather information on the location of road accident, the difference between the altitude of the accident point and that of the surrounding area, and the presence or absence of tunnels and bridges. A logistic regression model, artificial neural network (ANN) model, XGBoost model, and random forest model were used to investigate the relationship between the accidents caused by weather. Accuracy, kappa and area under the ROC curve (AUC) were used for the evaluation of machine learning models. Finally, a service model was proposed for road-level weather risk during the winter season. By crawling KMA database, weather information was collected and processed at the road level, and the risk of road accidents due to weather was predicted via machine learning using the constructed geometry and road environment information. Furthermore, through visualization of the predicted results on the map, a service that can be easily used by road users and decision makers in road management, was provided.

## 2. Data

### 2.1. Road Accident Data

The MOLIT OpenAPI Service presents the history of communication information, variable message sign (VMS) information, and construction/accident information. The accident information data consist of the time of accident, accident ID, accident characteristics,

latitude, and longitude. Based on this information, the accident characteristics are categorized into general road accidents, construction sections, and weather-related road accidents. As this study was conducted to investigate road accidents caused by inclement weather in winter, highway accident data were collected during the winter season from 2017 to 2019 (December, January, and February). And only road accidents that occurred within 200 m of the highway were considered in this study. The number of weather data/accident data on highways during the study period was 8683, out of which 8513 were general road accidents and 170 were weather-related.

### 2.2. Weather Data

To differentiate between weather-related and general accidents, road-level weather information is required. However, the KMA does not collect weather information based on road units. Road-level weather information can be estimated using spatial interpolation. Kim et al. proposed an optimal spatial model for each meteorological factor, using the weather observation data with the information for a distance between the highway and weather station within 1.5 km as test data, and farther than 1.5 km from the highway as training data. For precipitation and air pressure, the random forest model demonstrated the best performance, and for humidity and wind speed, the generalized additive model showed the best performance [14]. In this study, the temperature, humidity, precipitation, air pressure, and wind speed at the road accident occurrence point were estimated at a road level using nationwide weather data at the time of the road accident occurrence and used in this study.

### 2.3. Highway Data

Highways in Korea are standardized with node links. In this case, the standard node links consist of nodes and links connecting the nodes, developed to represent the road traffic network [15]. The link types of standard node links include general roads, bridges, tunnels, and overpasses, and link-level GIS information [16]. A total of 11,368 highway links were used in this study.

Furthermore, road geometry information of highways is needed to evaluate road accident risks. The geometric information used in the study was the angle of rotation and the turning radius of the road. The link is composed of many dots. For simplification to reduce the number of points, GIS and the Douglas–Peucker algorithm were used to generate the angle of rotation and turning radius of the road. Here, the Douglas-Peucker algorithm proposed by Douglas and Peucker is a method of removing unnecessary points by setting the threshold $\varepsilon$. On a straight line connecting the start and end points of the road, the vertical distance between the points is calculated, and the point with a vertical distance smaller than the set threshold is removed [17]. Because this algorithm depends on the distance between points and lines, it has the advantage of not being subject to constraints of dimensionality [18]. In this study, the highways were simplified by setting the threshold to 10 m.

Equation (1) shows the calculation method for the angle of rotation using points and lines in each link.

$$\theta = \arctan((y_2 - y)/(x_2 - x)) \times 180/\pi - \arctan((y_1 - y)/(x_1 - x)) \times 180/\pi \quad (1)$$

As a circle can be drawn with three points, it is drawn by selecting three points for each link from the standard node link data, and each geometric information calculated. Figure 1 shows the calculation method for geometric information.

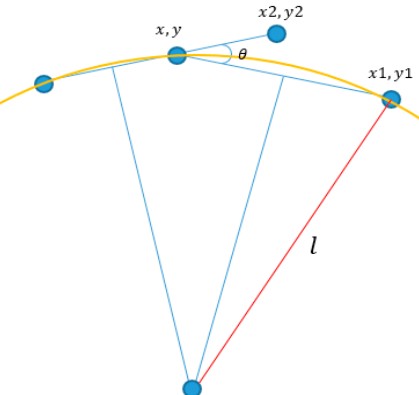

**Figure 1.** Calculation of angle of rotation and turning radius.

In the figure, $\theta$ represents the angle of rotation, and $l$ is the turning radius.

### 2.4. Shuttle Radar Topography Mission Data

Shuttle radar topography mission (SRTM) data are 3D digital elevation model data observed by satellites operated by NASA in 2000 [19]. They provide information on latitude, longitude, and altitude, with a resolution of 30 to 90 m. As the temperature is affected by altitude, this study used SRTM data with 30 m resolution to determine the altitude value of the area surrounding the road accident points and weather event occurrence. In addition, the difference between the altitude of the surrounding area and point of occurrence was used. Furthermore, as the duration of road freezing is prolonged owing to shadows, shadow information was generated using the time information and altitude difference values.

### 2.5. Construction of Combined Data

For training and testing of the machine learning models and logistic regression model, the combined data were divided into training data and test data in a ratio of 7:3. Table 1 shows the structure of the combined data. The training data applied to model the classification generally, but the data imbalance problem in the causes of road accidents must be resolved. As the ratio of general accidents was higher and the uncertainty of the model could be increased, the ratio of general road accidents to the weather-related road accidents in the training data was set to 1:1 using resampling technique.

To address the data imbalance problem, the methods are primarily classified into two categories. The first is, undersampling, which generates data of majority class with 1:1 ratio with reference to data in the minority class, and the other method is oversampling, which performs random sampling with replacement in the minority class with reference to the majority class at a 1:1 ratio [20]. The undersampling technique may remove important variables affecting the data, and the general oversampling technique duplicates the same data, which may lead to increased uncertainties in achieving high accuracy. Therefore, in this study, the data imbalance problem was resolved using the SMOTE method, based on the k-nearest neighbor algorithm [21], known to be more effective in improving the prediction accuracy of machine learning algorithms [22]. Total number of road accidents were 6079 with 5960 general road accidents, and 119 weather-related road accidents. Weather-related road accidents 6307 were resampled by the SMOTE algorithm corresponded to 6188 general road accidents to balance the data. Test data to verify the performance of the predictive model without resampling technique was used. The DMwR package of the R program was used to create a total of 10 databases with different seeds to solve imbalance problem [23].

**Table 1.** Structure of the combined data.

| No | Source | Lon. | Lat. | Accident | Road_Type | Max_Angle | Min_Radius |
|---|---|---|---|---|---|---|---|
| 1 | | 128.686 | 35.869 | Weather | Road | 24.864 | 56.889 |
| 2 | | 127.109 | 37.614 | Accident | Road | 32.075 | 42.691 |
| 3 | MOLIT &NodeLink | 126.820 | 37.584 | Accident | Bridge | 11.396 | 14.961 |
| ⋮ | | ⋮ | ⋮ | ⋮ | ⋮ | ⋮ | ⋮ |
| 8683 | | 126.634 | 37.46475 | Accident | Tunnel | 9.300 | 147.142 |

| No | Source | Temp | Ws | Prec | Hpa | Rh | |
|---|---|---|---|---|---|---|---|
| 1 | | 3.3 | 0.115 | 0 | 1019.6 | 77.338 | |
| 2 | | −2.5 | 3.116 | 0 | 1016.2 | 52.477 | |
| | KMA | 1.0 | 1.383 | 0 | 1027.4 | 73.974 | |
| ⋮ | | ⋮ | ⋮ | ⋮ | ⋮ | ⋮ | |
| 8683 | | 4.1 | 1.125 | 0 | 1027.8 | 56.885 | |

| No | Source | Altitude | diff_north | diff_south | diff_west | diff_east | cov_west | cov_east |
|---|---|---|---|---|---|---|---|---|
| 1 | | 39 | 1 | 1 | −1 | 1 | 0 | 0 |
| 2 | | 38 | 1 | −1 | 1 | 0 | 0 | 0 |
| | SRTM | 10 | −4 | 0 | −28 | 0 | 0 | 28 |
| ⋮ | | ⋮ | ⋮ | ⋮ | ⋮ | ⋮ | ⋮ | ⋮ |
| 8683 | | 9 | −1 | 1 | −1 | −7 | 0 | 11 |

## 3. Methods

### 3.1. Logistic Regression Model

A general regression model finds a linear relationship between an independent variable and a dependent variable. The logistic regression model is similar in that it explains the dependent variables expressed as 0 and 1 using the linear connection of the independent variables [24]. However, the logistic regression model predicts the probability $\pi(x)$ by which the dependent variable falls under the class of interest in terms of the linear function g(x) of the explanatory variable. The logistic regression model is presented as follows [25]:

$$\pi(x) = \frac{exp[g(x)]}{1 + exp[g(x)]}, \quad g(x) = \ln\left[\frac{\pi(x)}{1 - \pi(x)}\right] = \beta_0 + \sum \beta_i x_i, \tag{2}$$

where $\beta_0$ is the intercept of the linear model, and $\beta_i$ is the regression coefficient of each variable. The regression coefficient was estimated using the glm function, a built-in function in the R Stats Package [26].

### 3.2. Neural Network Model

The artificial neural network (ANN) model uses a linear classifier with a perceptron, which was first proposed by Rosenblatt and is an algorithm that receives multiple signal inputs and produces one signal output. This machine-learning algorithm is inspired by biological neurons [27,28]. In this study, the ANN model was created using the nnet function built into the R nnet package, and the hyper-parameters used were the number of hidden layer nodes (size) and a weight to prevent overfitting (decay) [29].

### 3.3. Random Forest Model

The random forest model, proposed by Breiman, is an advanced technique to improve the predictive performance and overfitting problems of the decision tree model, and is

an ensemble technique that uses resampling-based bagging [30]. Here, multiple data are constructed by bootstrap sampling, and a decision tree grows from each bootstrap data with a limited number of variables. In the process of aggregating the prediction results of the decision tree, the majority vote is used for classification purposes, and the average of each tree is used for regression problems [31]. In this study, a model was created using the randomForest function built in the randomForest package of R, and the hyper-parameter used for the setting, was the number of explanatory variables (mtry) [32]. The number of trees was set to 500.

### 3.4. eXtreme Gradient Boosting (XGBoost)

XGBoost is an ensemble technique that uses a boosting algorithm to perform training to increase predictive performance by applying continuous weight updates to learning results [33]. It is a model with weight restriction, to improve the problem of the absence of overfitting regulation and execution time of gradient boosting (GBM) and has a fast learning speed and excellent performance [34]. This study used xgbtree, a built-in function in the xgboost package of R [35]. The hyper-parameters used in the study were as follows: number of iterations(controlling the maximum number of iterations)(nrounds), maximum allowable depth(the maximum depth of a tree)(max_depth), reduction of loss function (minimum loss reduction required to make a further partition on a leaf node of the tree)(gamma), minimum number of instances in a child node (minimum sum of instance weight, needed in a child node)(min_child_weight), subsample ratio of the training instance(subsample), subsample ratio of columns when constructing each tree(colsample_bytree), and weight for each training step (step size shrinkage used in update to prevent overfitting)(eta).

### 3.5. k-Fold Cross Validation

In the k-fold cross validation proposed by Geisser, the original sample with SMOTE algorithm is randomly partitioned into k equal-sized subsamples and among the k subsamples, a single subsample is retained as the validation data for testing the model, and the remaining k-1 subsamples are used as training data. This cross-validation process is iterated k times [36]. The k value used in this study was 5. This technique was used to obtain the optimal hyper-parameters for 10 randomly generated training data. To tune the hyperparameters, a random search was performed 500 times.

### 3.6. Performance Evaluation

In this study, accuracy, kappa, and AUC were used as metrics to evaluate the prediction performance of the prediction models. In addition, test data were evaluated using F-measure and G-mean additionally [37]. A confusion matrix presents the agreement between the actual observed data and the predicted value [38]. Table 2 shows the confusion matrix

**Table 2.** Confusion matrix.

|  |  | Predicted Class | |
|---|---|---|---|
|  |  | **True** | **False** |
| Actual | True | True Positive(TP) | False Negative(FN) |
| Class | False | False Positive(FP) | True Negative(TN) |

Here, True Positive and True Negative, are cases wherein the observed data and the predicted values are in agreement, and False Negatives and False Positives are cases wherein the actual data and predicted values are not in agreement. Accuracy and Kappa are calculated by the following equation using the confusion matrix:

$$Accuracy = \frac{TP+TN}{TP+TN+FP+FN}, \quad Kappa = \frac{Accuracy-P}{1-P},$$

$$F-measure = \frac{2 \times \frac{TP}{TP+FP} + \frac{TP}{TP+FN}}{\frac{TP}{TP+FP} + \frac{TP}{TP+FN}}, \quad G-mean = \sqrt{\frac{TP}{TP+FN} \times \frac{TN}{TN+FP}} \quad (3)$$

where *P* is the ratio of agreement between the actual observed data and the predicted value in random sampling [39].

## 4. Results

Table 3 shows the mean and standard deviation of the validation prediction performance for the optimal hyper-parameters obtained by a 5-fold cross validation on the training data with resampling technique. Results of performance evaluation of data with resolved imbalance, the random forest model achieved an accuracy = 0.991, kappa = 0.982, and AUC= 0.986, which showed its superior prediction performance compared to the other models. When evaluated based on accuracy, the random forest model showed the highest accuracy, followed by XGBoost, neural network, and logistic regression. The optimal hyper-parameters selected for each model were as follows: random forest (mtry = 2), XGBoost (eta = 0.272, max_depth = 8, gamma = 1.351, colsample_bytree = 0.341, min_child_weight = 1, subsample = 0.824, nrounds = 455), and neural network = 0.975 (size = 20, decay = 0.057).

**Table 3.** The prediction performance of validation data of balance data.

| | N | Accuracy | | Kappa | | Auc | |
|---|---|---|---|---|---|---|---|
| | | Mean | SD | Mean | SD | Mean | SD |
| Logistic | 10 | 0.823 | 0.003 | 0.645 | 0.005 | 0.691 | 0.057 |
| Neural Network | 10 | 0.972 | 0.001 | 0.945 | 0.003 | 0.958 | 0.012 |
| XGBoost | 10 | 0.991 | 0.001 | 0.982 | 0.002 | 0.978 | 0.008 |
| Random Forest | 10 | 0.991 | 0.001 | 0.982 | 0.002 | 0.986 | 0.002 |

As the random forest model and the XGBoost model use algorithms based on CART, variables that significantly reduce impurities can be identified [40]. Figure 2 shows the calculated average of the variable importance of the 10 training data generated by the SMOTE technique. Geographic location (lon, lat) and humidity (rh) were identified as important factors in both random forest and XGBoost. Thus, these are the most important factors in road accidents caused by weather. When high humidity combines with low temperature, fog and frost can occur, leading to the accumulation of ice on the roads, and this has a significant impact on weather-related accidents in the winter season [39]. Among the various types of weather information, temperature, wind speed, and air pressure were identified as important factors in road accidents caused by weather, and precipitation was relatively insignificant. Among the road geometry factors, roads with a short turning radius and a large angle of rotation were identified as the main causes of road accidents due to weather, and the presence or absence of tunnels and bridges was not significant. Among the road environment variables, altitude demonstrated the greatest effect on road accidents due to weather, and the difference in altitude from the surrounding area had an effect in the order of north, south, west, and east side, of the surrounding area.

Table 4 shows the summary of the logistic regression model. 'Est.' denotes estimates of logistic regression model and Exp(Est.) denotes exponential estimates which means odds ratio. A 95% confidence interval (CI) shows the significance of the independent variables. The statistically significant variables of the logistic regression model are hour, latitude, and relative humidity. Hour and latitude have a negative relationship and relative humidity has a positive relationship with weather-related accidents. The results were similar to the order of important variables derived from machine learning.

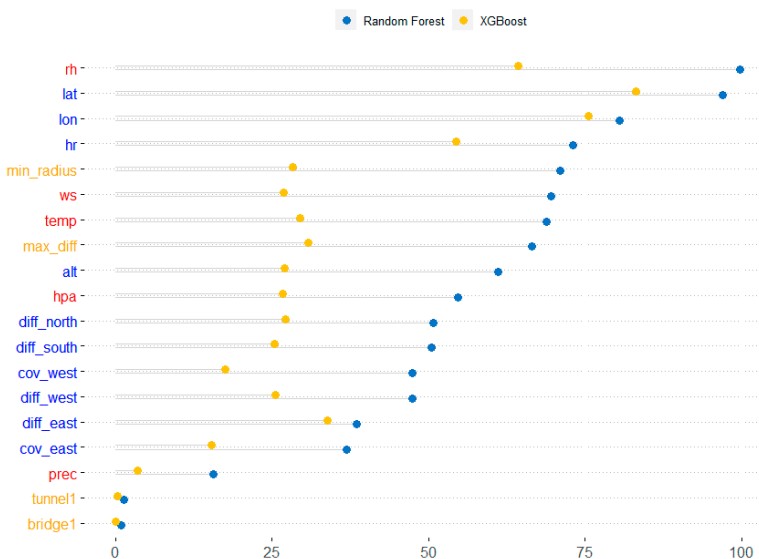

**Figure 2.** Important variables of Random Forest and XGBoost.

**Table 4.** The summary of logistic regression model.

| Variable | Est. | Exp(Est.) | 95% CI Lower | 95% CI Upper | z |
|---|---|---|---|---|---|
| hr | −0.089 | 0.915 | −0.129 | −0.049 | −4.371 *** |
| lon | −0.140 | 0.870 | −0.433 | 0.154 | −0.932 |
| lat | −0.986 | 0.373 | −1.244 | −0.729 | −7.500 *** |
| alt | 0.004 | 1.004 | −0.001 | 0.009 | 1.700 |
| diff_up | 0.005 | 1.005 | −0.036 | 0.046 | 0.245 |
| diff_down | 0.009 | 1.009 | −0.037 | 0.056 | 0.389 |
| diff_right | −0.007 | 0.993 | −0.020 | 0.007 | −0.971 |
| diff_left | 0.003 | 1.003 | −0.011 | 0.018 | 0.420 |
| temp | 0.038 | 1.039 | −0.006 | 0.082 | 1.692 |
| ws | 0.207 | 1.230 | −0.004 | 0.418 | 1.924 |
| prec | −0.063 | 0.938 | −0.408 | 0.281 | −0.362 |
| rh | 0.051 | 1.052 | 0.038 | 0.064 | 7.827 *** |
| hpa | −0.001 | 0.999 | −0.043 | 0.040 | −0.071 |
| min_radius | 0.000 | 1.000 | 0.000 | 0.000 | −0.039 |
| max_diff | −0.017 | 0.984 | −0.037 | 0.004 | −1.557 |
| cov_right | 0.000 | 1.000 | −0.001 | 0.001 | 0.778 |
| cov_left | 0.000 | 1.000 | −0.001 | 0.001 | −0.353 |
| tunnel | 0.398 | 1.488 | −1.701 | 2.497 | 0.371 |
| bridge | −0.031 | 0.970 | −2.078 | 2.017 | −0.029 |

*** $p < 0.001$, ** $p < 0.01$, * $p < 0.05$.

Table 5 presents the results of the prediction performance evaluation of the test data for different models using the optimal hyper-parameters. The random forest model demonstrated the highest prediction performance with an accuracy = 0.984, kappa = 0.658, and AUC = 0.907, F-measure = 0.992, G-mean = 0.904 which was far greater than that of the other models. The prediction performance, in increasing order of accuracy was, the random forest model, XGBoost, neural network, and logistic regression.

**Table 5.** The prediction performance of test data.

| | N | Accuracy | Kappa | AUC | F-measure | G-mean |
|---|---|---|---|---|---|---|
| | | Mean(SD) | Mean(SD) | Mean(SD) | Mean(SD) | Mean(SD) |
| Logistic | 10 | 0.742 (0.078) | 0.085 (0.023) | 0.790 (0.024) | 0.847 (0.056) | 0.786 (0.030) |
| Neural Network | 10 | 0.940 (0.008) | 0.331 (0.030) | 0.879 (0.008) | 0.969 (0.004) | 0.877 (0.009) |
| XGBoost | 10 | 0.980 (0.003) | 0.613 (0.038) | 0.898 (0.010) | 0.990 (0.002) | 0.894 (0.012) |
| Random Forest | 10 | 0.984 (0.001) | 0.658 (0.019) | 0.907 (0.008) | 0.992 (0.001) | 0.904 (0.008) |

The random forest model showed the best prediction performance for weather-related warnings and road accidents. The result of XGBoost demonstrated similar performance to the random forest in the training data, but as the random forest shows superior performance in the test data, it can be considered that overfitting occurred in the training data of XGBoost.

## 5. Visualization Service

If the risk of accidents due to weather can be visualized on the map in real time on a road level, it would prove useful for enhancing the road safety in the winters. To predict traffic accidents caused by weather, information on road geometry of highways in Korea, road environment, and weather is needed. In this case, information on road geometry of highways and road environment comprises fixed values and is obtained using geographic information system (GIS) information of standard node links and the SRTM data. There were a total of 300,914 GIS links and 11,368 links on highways. To reduce the computation time, a representative GIS for each link was used. For the representative GIS information, the location of past road accidents caused by weather and the road geometry information comprising the angle of rotation and the minimum radius obtained by calculating the sum of the squares of the distance between the radii, is needed. As the accident risk is higher with an increasing angle of rotation, the angle of rotation and turning radius of GIS with the largest angle of rotation were used. Furthermore, road environment information was obtained based on this point. Weather information constantly changes. Therefore, weather information data need to be crawled at given time, and spatial interpolation should be performed. Weather data provided by KMA were collected using the readHTMLtable function of the RCurl package [41] of R, and among the collected weather data, road-level information of precipitation, air pressure, and temperature were obtained by the random forest model, and humidity and road-level information of wind speed were obtained using the generalized additive model. If the generated road geometry, road environment, and weather data are applied as input variables to the random forest model, which is the risk prediction model for weather-related accidents, the occurrence of road accidents due to toad-level weather conditions can be predicted. Finally, visualization of the status of weather-related road accidents with a leaflet [42], an open map, is significantly useful for road safety management in the winter season.

On 7 January 2021, the temperature was below zero Celsius nationwide, and snowfall occurred in the metropolitan area and southern regions. Figure 3 shows a visualization of the humidity information, which has the most significant impact on Korean highway information along with the angle of rotation, and weather information. Humidity was high in the western and southern regions of the Korean Peninsula.

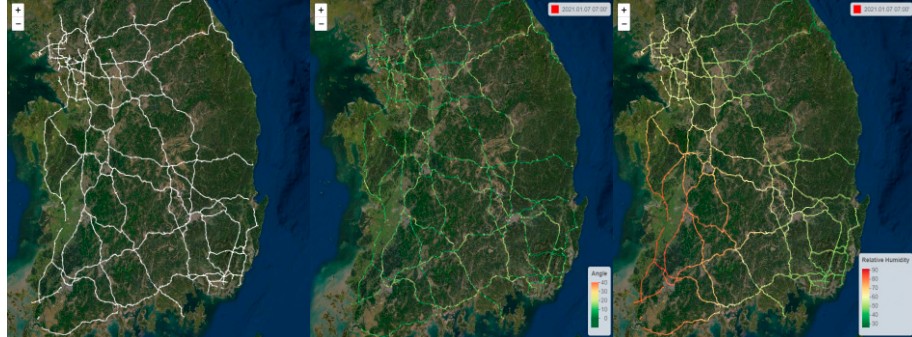

**Figure 3.** Location of Korean highways, angle of rotation by link, humidity by link.

Figure 4 shows a visualization of the prediction results of road accidents due to weather conditions on 7 January 2021, based on road geometry information of highways, road environment, and weather information. Highways in the southwestern region were identified as high risk for weather-related road accidents. The number of links selected as points of risk was 73, and highways with many points of risk were the West Coast Expressway and the Gwangju-Daegu Expressway. In this study, the visualization service was presented as an example based on 7 January 2021, but it can be implemented in real time, so that it can practically guide and alert users on the road to the risk of road accidents due to inclement weather. In addition, if the time point of the previous road accident is entered, a researcher can check the road environment, road geometry, and weather information of the road accident point, so it has the advantage of examining the causes of road accident from multiple perspectives.

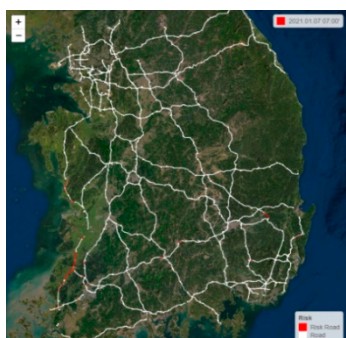

**Figure 4.** Risk of road accidents on highways caused by weather in Korea.

## 6. Conclusions

This study evaluated the risks of traffic accidents on highways in winter by classifying accidents into those caused specifically by the weather and general road accidents. As the weather information on the highways could not be directly obtained, the wind speed and humidity were predicted using the road-level weather data by spatial interpolation of the data of the surrounding weather stations. To determine the risk of road accidents on highways, the angle of rotation and turning radius were calculated using the Douglas–Peucker algorithm and standard node links, and the road with the largest angle of rotation was selected as the representative point of each link. Using SRTM data, the difference between road altitude and that of the surrounding area and shadow information were produced to construct the combined data. To verify the prediction performance of the model, the combined data were divided into training data and test data in a 7:3 ratio. The data imbalance problem was resolved using the SMOTE algorithm for the training data, and 10 combined databases were constructed. Random forest, XGBoost, ANN model, and logistic regression model were used as machine learning models to predict the risk of accidents due to weather, and random forest was selected as the optimal model

(accuracy = 0.984, kappa = 0.658, AUC = 0.907, F-meaure = 0.992, G-mean = 0.904). As a result of identifying important factors through the random forest model and the XGBoost model among the CART-based models, the important variables for predicting road accident risks due to weather were identified to be (in increasing order of importance) humidity, geographic location (latitude, longitude), time, and road geometry information. Finally, the visualization service enabled an easy understanding of the conditions for the road users and researchers, by means of various information on road accidents in the winter season.

This study had several limitations. This study predicted weather information using spatial interpolation because it was not possible to collect direct weather information on highways. As accurate road weather information cannot be obtained, there is uncertainty in the accuracy of predicting accidents caused by weather. Accidents due to the slipperiness of the road are affected by road material, road traffic volume, driving speed, road slope, and road surface temperature. Therefore, research on winter road risk prediction reflecting these various factors remains a future challenge.

**Author Contributions:** D.K.: methodology, visualization, analysis, writing; S.J.: collecting data, methodology; S.Y.: conceptualization, writing original draft preparation, review and editing, supervision. All authors have read and agreed to the published version of the manuscript.

**Funding:** This research was supported by Daegu University Future Scholars program, 2021.

**Institutional Review Board Statement:** Not applicable.

**Informed Consent Statement:** Not applicable.

**Data Availability Statement:** The data presented in this study are available upon request from the corresponding author.

**Acknowledgments:** The authors would like to express their gratitude to all those who contributed to this study.

**Conflicts of Interest:** The authors declare no conflict of interest.

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
