# Peer review of "Risk Prediction for Winter Road Accidents on Expressways"

_applsci, doi:10.3390/app11209534_

Round 1

Reviewer 1 Report

If I have understood correctly, the classification of the accident type into other and those caused by weather is already included in the MOLIT source data. It is unclear, however, according to which criteria this classification was made. This seems to me to be very important in order to be able to interpret the results. 

Reviewer 2 Report

The paper is very interesting, but lacks methodological rigor:

  • The Authors use SMOTE for an imbalance problem, in this world they create a new dataset. The authors must insert summary statistics on the accident data on the 2 datasets, training data and test data, both in the original version and after resampling with SMOTE.
  • It is not clear if SMOTE was applied to the 2 datasets, training data and test data, or only to the training set.
  • The Authors write "The training data can be applied to the machine learning model". It is unclear whether it was also applied to the logit model.
  • The authors use the logit model without considering any autocorrelation, see “Mannering, F. L., Shankar, V., & Bhat, C. R. (2016). Unobserved heterogeneity and the statistical analysis of highway accident data. Analytic methods in accident research11, 1-16”.
  • - The authors write “In the k-fold cross validation proposed by Geisser, the original sample is randomly partitioned into…”. It is therefore not clear for which technique it was used and if therefore the new dataset created with SMOTE was used only for some models.
  • - The performance measures used are not very suitable when applying balancing methods, more suitable metrics are F-Measure adn G-mean (Liu, XY, Wu, J., Zhou, ZH, (2009). Exploratory Under Sampling for Class Imbalance Learning, IEEE Transactions on Systems, Man, and Cybernetics — Part B: Cybernetics, VOL. 39 (2); Chen, C., A., Liaw, L., Breiman, 2004. Using random forests to learn unbalanced data. Technical Report 666, Statistics Department, University of California at Berkeley).
  • - The results are incomplete. Authors should insert a table with the coefficients of the significant logit models. Authors should insert a summary table with the impacts of individual variables for different similar elasticity models see Washington, S., Karlaftis, M., Mannering, F., & Anastasopoulos, P. (2020). Statistical and econometric methods for transportation data analysis. Chapman and Hall / CRC.

Round 2

Reviewer 2 Report

The authors responded to all reviews.